# Association of sarcopenia with important health conditions among community-dwelling Asian women

**Beverly Wen-Xin Wong**[1], **Win Pa Pa Thu**[1], **Yiong Huak Chan**[2], **Susan Jane Sinclair Logan**[1], **Jane A. Cauley**[3], **Eu-Leong Yong**[1]*

**1** Department of Obstetrics and Gynecology, National University Hospital, National University of Singapore, Singapore, Singapore, **2** Biostatistics Unit, Yong Loo Lin School of Medicine, National University of Singapore, Singapore, Singapore, **3** Department of Epidemiology, Graduate School of Public Health, University of Pittsburgh, Pittsburgh, PA, United States of America

* obgyel@nus.edu.sg

## Abstract

This study aimed to examine sarcopenia prevalence using the Asian Working Group for Sarcopenia 2019 (AWGS) and the Foundation for the National Institutes of Health (FNIH) definitions, and their associations with important health conditions affecting midlife Singaporean women. Muscle mass and function were objectively assessed in 1201 healthy community-dwelling subjects aged 45–69 years under the Integrated Women's Health Program (IWHP). Dual-energy X-ray absorptiometry (DXA), handgrip strength and the Short Physical Performance Battery (SPPB) were measured, and the relationship between sarcopenia with hypertension, type 2 diabetes (T2DM), osteoporosis, depression/anxiety, and urinary incontinence were examined using binary logistic regression models. Sarcopenia prevalence was 18.0% and 7.7% by the AWGS and FNIH criteria respectively. Osteoporosis (aOR: 1.74, 95% CI: 1.02, 2.94) and T2DM (aOR: 1.98, 95% CI: 1.14, 3.42) was positively associated with AWGS- and FNIH-defined sarcopenia respectively, while hypertension was not, after adjustment for age, ethnicity, education levels and menopausal status. A negative percent agreement of 95.6% suggests good agreement between the criteria in the absence of sarcopenia. Even though they represent a single concept, sarcopenia by either criterion differed in their relationships with diabetes and osteoporosis, suggesting the need for further rationalization of diagnostic criteria.

## Introduction

Sarcopenia, the loss of skeletal muscle mass with advancing years, is associated with increased risk of falls [1], functional decline, reduced quality of life [2], higher hospitalization rates and related costs [3], cardiovascular diseases [4]), disability [5], and even mortality [5, 6]. The European Working Group on Sarcopenia in Older People formulated a definition of sarcopenia in 2010 [7], which was revised in 2018 to focus on muscle strength [8]. Based on the European definition, the Asian Working Group for Sarcopenia's definition in 2019 (AWGS)

**Funding:** This study was funded by the Singapore National Medical Research Council Grant (Reference number: NMRC/CSASI/0010/2017) for ELY. There was no additional external funding received for this study.

**Competing interests:** The authors report that there are no competing interests to declare.

accounted for differences in anthropometry, lifestyle, and culture [9]. Threshold cut-offs relied on distributional norms with reference to a young adult population. Another commonly used definition was developed by the Foundation for the National Institutes of Health (FNIH) in 2014. Using the same criteria of muscle mass and strength, the FNIH criteria was framed conceptually to discriminate mobility impairment [10], and was validated with a large dataset [11]. Although a lot of progress has been made, a universally accepted definition of sarcopenia remains elusive.

In Singapore, 30.9% of women (aged ≥ 60 years) were sarcopenic using the AWGS definition [12]. Despite this high sarcopenia prevalence, only small-scale studies have been conducted locally using varying sarcopenic definitions [12–15]. Chew et al. defined sarcopenia solely based on muscle strength [13]; Low et al. defined sarcopenia based on skeletal muscle mass alone [15], while other studies defined sarcopenia based on a mix of muscle mass, strength and/or performance [12–14], resulting in controversy on the prevalence of sarcopenia.

The main risk factor for sarcopenia is aging. As such, studies on sarcopenia were mostly conducted in elderly subjects [8]. Nevertheless there is increasing realization that the development of sarcopenia may start earlier in life [16], emphasizing the need to study sarcopenia across the life course. Women have lower absolute muscle mass than men [17], and there is some evidence that muscle mass reduction accelerates during the time around menopause [18]. Decrease in estrogen levels post-menopause have been associated with a decrease in muscle mass and function [18, 19]. Hence, studies on sarcopenia should also focus on younger women as much as in geriatric men. In addition, most studies globally [8, 9] and in Singapore [14, 15] focused mainly on the relationship between sarcopenia and single health conditions. In this respect, the Integrated Women's Health Program (IWHP) has identified important health conditions that manifest in midlife women, including high systolic blood pressure [20], insulin resistance [21], osteoporosis [22], depression and anxiety [23], and urinary incontinence [24]. Since hypertension, T2DM, osteoporosis, depression and anxiety, and urinary incontinence are the most common and critical health conditions, we studied their association with sarcopenia as these conditions are likely to progress and result in a higher healthcare burden on midlife women as they age. Identification of the link between sarcopenia and these common health conditions would allow for specific preventive healthcare initiatives.

Our study aims to firstly, examine the prevalence of sarcopenia among midlife Singaporean women using two recently updated definitions (AWGS and FNIH). Secondly, we examined the associations of hypertension, type 2 diabetes (T2DM), osteoporosis, depression and anxiety, and urinary incontinence with sarcopenia.

## Materials and methods

### Study design and participants

The IWHP is a cross-sectional study examining key health issues among midlife Singaporean women [25]. The cohort has been fully described previously [25]. Briefly, healthy women aged 45–69 years receiving routine gynecological follow-up at well-women clinics at the National University Hospital were recruited from September 2014 to October 2016 through fliers, posters, and word-of-mouth. Exclusion criteria included pregnancy or being severely ill. The number of women screened was at 2715, and 2191 met the eligibility criteria. Thereafter, 746 subjects, 134 subjects and 110 subjects declined participation, failed to attend the appointment and were non-contactable respectively, leaving a final sample of 1201 women [25]. The protocol was approved by the Domain Specific Review Board of the National Healthcare Group, Singapore (Reference number: 2014/00356) and all participants provided written informed

consent. In short, a whole-body composition scan was performed, and a fasting blood sample was collected for participants on arrival. This was followed by light refreshments, a series of questionnaires, biophysical measurements, and physical function tests, with the visit approximating 90 minutes. Participants were also asked to bring all prescriptive, over the counter, traditional medications, and supplements consumed in the past two weeks to the study visit. The names and dosages of all medications and supplements were then recorded by study coordinators [25]. This study was performed in concordance with all the items required in the STROBE statement [26].

## Health conditions

The five important health conditions assessed were hypertension, T2DM, osteoporosis at the lumbar spine, depression and anxiety, and urinary incontinence. Blood pressure was measured thrice while in a seated position using the OMRON Intellisense device (HEM7211), with the average calculated. Hypertension was defined as systolic blood pressure ≥140 mmHg and/or diastolic blood pressure ≥90mmHg or use of anti-hypertensive drugs or self-reported physician-diagnosed hypertension via a questionnaire. Blood was drawn following an overnight fast. Fasting blood glucose levels were obtained. Details on the concentration ranges and coefficients of variance are described in **S1 Table**. Participants were classified as diabetic if they had fasting blood glucose levels ≥ 7.0 mmol/l, or were on anti-diabetic medications, or self-reported physician-diagnosed diabetes via a questionnaire. Bone mineral density at the lumbar spine was obtained using dual-energy X-ray absorptiometry (DXA). Participants were classified into non-osteoporotic (T-score >-2.5) and osteoporotic (T-score ≤ −2.5) at the lumbar spine in accordance to the World Health Organization's guidelines [27]. Calculation of T-scores were based on reference data from Singaporean women.

Depression symptoms were assessed using the Center for Epidemiological Studies for Depression Scale, which comprised of a 20-item scale ranging from a score of 0 to 60 [28]. Anxiety symptoms were assessed using the General Anxiety Disorder Scale [29]. Women were classified to have anxiety and/or depression symptoms if they had a score of ≥16 using the depression scale and/or a score of ≥10 using the anxiety scale or if they reported use of anti-anxiety or anti-depressive medications. Urinary incontinence was determined from a subscale of the Pelvic Floor Disability Index, that comprises of 20 questions and 3 subscales, and has been validated and widely used to assess women with pelvic floor disorders [30]. Women were classified as having urinary incontinence if they had any type of incontinence (stress, urge, mixed or leakage (drops) only). Details on how the different types of incontinence were defined, as well as the scoring system, have been previously published [24].

## Sarcopenia parameters and definitions

Appendicular lean mass (ALM) was measured using DXA (Hologic Discovery Wi, Apex software 4.5) [25]. Operators of the DXA machine were trained to follow standard protocols according to the manufacturer's instructions. The DXA machine was calibrated daily and sent for maintenance every 6 months. Handgrip strength was assessed using a dynamometer (Jamar, Bolingbrook, IL), which was calibrated yearly. Participants held the dynamometer in a seated position, with their elbows flexed at a 90˚ angle and with the forearm parallel to the ground, squeezing as hard as they could. Two measurements were taken each on both arms, and the maximum value out of four readings was analyzed. The Short Physical Performance Battery (SPPB) assessed lower extremity strength, static balance and gait speed [31]. Lower extremity strength was evaluated by the repeated chair stand test, which recorded the time taken for participants to rise five times from a chair, unaided. Static balance was assessed by

ability to perform the semi-tandem, tandem and one-leg stands for 30 seconds. Gait speed (m/s) was assessed via usual and fast narrow walks, whereby participants walked 6m within two narrow lines 20cm apart. The SPPB has a maximum score of 12 points, with $\geq 10$ indicating high performance and $\leq 9$ indicating moderate to low performance [31].

Participants were classified as having sarcopenia according to the following definitions:

1. AWGS: appendicular lean mass/height$^2$ (ALM/height$^2$) $<5.4$ kg/m$^2$ together with handgrip strength $< 18$ kg, or SPPB score $\leq 9$ points [9].

2. FNIH: ALM/body mass index (BMI) $<0.512$ and handgrip strength $<16$ kg [32].

## Other variables

Age, ethnicity, education level, menopausal status, smoking status, and alcohol consumption were self-reported using a questionnaire. Regular exercise was defined as having spent at least 150 minutes per week on moderate intensity physical activity and/or 75 minutes per week on vigorous intensity physical activity, measured by the Global Physical Activity Questionnaire [33]. Total serum 25-hydroxyvitamin D was measured as the sum of metabolites, 25(OH)D$_2$ and 25(OH)D$_3$, using liquid chromatography-tandem mass spectrometry [34]. Participants were classified into either $\leq 20$ ng/ml (vitamin D-deficient) or $> 20$ ng/ml (normal vitamin D levels) [35]. Visceral adipose tissue (VAT) was identified by automated software algorithms from DXA, was analyzed in cm$^2$ and classified into tertiles: lower ($<88.6$ cm$^2$), middle (88.6–131 cm$^2$) and upper ($> 131$cm$^2$). The bottom two tertiles were combined, and VAT was then presented as a dichotomous variable, $\leq 131$ cm$^2$ and $> 131$ cm$^2$. hs-CRP, IL-6, and TNF-$\alpha$ inflammatory markers were obtained from fasting blood samples and measured using ELISA assay kits (DRG International, Inc. USA), chemiluminescence immunoassays (ADVIA Centaur Analyzer, Siemens Healthcare Diagnostics) and Colorimetric (Beckman Coulter, Inc., USA) respectively. They were presented as tertiles: (IL-6 as $<1.4$, 1.41–2.5, or $>2.5$ pg/mL; TNF-$\alpha$ as $<5.4$, 5.5–7.3, or $>7.3$ pg/mL; hs-CRP as $<0.7$, 0.71–1.8, or $>1.8$ mg/L). Assay parameters are detailed in **S1 Table**.

## Statistical analysis

Univariate analyses examined the relationship between participant characteristics and sarcopenia using one-way ANOVA and Pearson's Chi-Square test. Continuous variables following a normal distribution were expressed as mean ± SD, while categorial variables were expressed as n (%). The normality of the variables was examined visually using histograms and Q-Q plots, as well as using the Kolmogorov-Smirnov and Shapiro-Wilk tests.

Comparison between the prevalence of sarcopenia by AWGS and FNIH definitions were performed using (i) positive percent agreement: the proportion of participants who were categorized as having the condition by both the FNIH and AWGS criteria divided by the number of participants who were categorized as having the condition by AWGS; (ii) negative percent agreement: the proportion of participants who were categorized as not having the condition by both the FNIH and AWGS criteria divided by the number of participants who were categorized as not having sarcopenia by AWGS; (iii) Cohen's $\kappa$ was calculated to determine the levels of agreement between the criteria. *Kappa* values $< 0.40$ indicate poor reliability, 0.40 to 0.75 indicate fair-to-good reliability and $> 0.75$ indicate excellent reliability [36, 37].

The five different health conditions were dichotomized, and we used binary logistic regression to explore the associations between each health condition and sarcopenia. We adjusted for demographic and lifestyle factors that had a p-value of $<0.05$ on univariate analysis. Age, ethnicity, education level and menopausal status were entered into Model 1. Model 1 and VAT

were entered into Model 2 to adjust for central obesity. VAT was substituted for BMI as the former has been found to provide additional information in terms of systemic inflammation [38], which is important in the mediation of various health conditions. The results were expressed as Odds Ratio (OR), adjusted Odds Ratio (aOR) and 95% Confidence Interval (CI).

All results were analyzed using SPSS software (Version 27.0, Chicago, IL, USA) and statistical significance was achieved at $p < 0.05$.

## Results

### Participant characteristics

The 1201 women had a mean age of 56.3 years. Chinese women made up the majority at 81.1% (**Table 1**), while Malay and Indian women constituted 5.5% and 9.9% respectively. The remaining 3.5% constituted of women from other ethnicities. Most women were postmenopausal (71.7%), while the vast majority do not smoke (97.4%) and do not drink alcohol (96.0%). Roughly two-thirds (60.8%) of women exercise regularly. Fifty women were defined as sarcopenic by both the FNIH and AWGS criteria (**Fig 1**). The positive percent agreement between the two definitions was low at 23.1%, while the negative percent agreement was high at 95.6%. Cohen's $\kappa$ of 0.24 (95%CI: 0.17, 0.31) indicated poor agreement between the two criteria.

The prevalence of sarcopenia by the AWGS criteria was 18.0% (**Table 1**). Sarcopenic subjects were older (mean age: 57.9 vs 55.9) and were more likely to be Chinese or Indian. They also reported lower education, postmenopausal status, and had lower VAT compared to non-sarcopenic women. They also had lower muscle mass, lower maximal grip strength, and reduced muscle function in the repeated chair stand, tandem and one-leg stand tests. However, they were faster in the usual and narrow gait speed tests compared to non-sarcopenic women.

The prevalence of sarcopenia by the FNIH definition was 7.7%. Sarcopenic subjects were older (mean age: 59.1 vs 56.1) and were more likely to be Malay or Indian. They also reported lower education and postmenopausal status. Unlike AWGS, sarcopenic women by FNIH had more VAT, and had higher proportions in the highest tertile of hs-CRP, IL-6, and TNF-α. Like AWGS, sarcopenic subjects have lower muscle mass, lower maximal grip strength, and reduced ability on the repeated chair stand, tandem and one-leg stand tests but were faster in the usual and narrow tests.

### Associations with diabetes and osteoporosis

The prevalence of key health conditions among the participants in descending order were hypertension (55.3%), urinary incontinence (52.3%), anxiety and/or depression symptoms (17.0%), T2DM (11.9%), and osteoporosis (6.3%).

Among those with osteoporosis, 30.3% were sarcopenic by the AWGS definition (**Table 1**). Women with osteoporosis were more likely to be sarcopenic (aOR:1.74, 95% CI: 1.02, 2.94), after adjustment for age, ethnicity, education levels and menopausal status (Model 1) (**Table 2**). However, this association was attenuated upon VAT adjustment. Hypertension, T2DM, depression and/or anxiety symptoms, and urinary incontinence were not significantly associated with AWGS sarcopenia.

Among those with hypertension, 9.9% were sarcopenic; and in women with diabetes, 15.5% were sarcopenic by the FNIH definition (**Table 1**). Hypertension was associated with 2.10-fold (95% CI: 1.31, 3.36) higher odds of sarcopenia, which was attenuated after adjustment for Model 1 (**Table 2**). Women with diabetes were 1.98 times (95% CI: 1.44, 3.42) more likely to be sarcopenic after adjustment for demographic factors in Model 1. Further adjustment for VAT (Model 2) resulted in the attenuation of this association. Osteoporosis, depression and/or anxiety symptoms, and urinary incontinence were not significantly associated with FNIH sarcopenia.

**Table 1. Participant characteristics (n = 1201).**

| Characteristics | Total n = 1201 | AWGS | | P-value | FNIH | | P-value |
|---|---|---|---|---|---|---|---|
| | | Sarcopenia n = 216 (18.0%) | No sarcopenia n = 985 (82.0%) | | Sarcopenia n = 93 (7.7%) | No sarcopenia n = 1108 (92.3%) | |
| **Demographics** | | Mean ± SD or N (%) | | | | | |
| Age (years) | 1201 | 57.9 ± 6.3 | 55.9 ± 6.1 | <0.001*** | 59.1 ± 6.2 | 56.1 ± 6.1 | <0.001*** |
| Ethnicity | 1159 | | | 0.031* | | | 0.003** |
| Chinese | 974 | 185 (19.0) | 789 (81.0) | | 65 (6.7) | 909 (93.3) | |
| Malay | 66 | 4 (6.1) | 62 (93.9) | | 9 (13.6) | 57 (86.4) | |
| Indian | 119 | 22 (18.5) | 97 (81.5) | | 17 (14.3) | 102 (85.7) | |
| Education level | 1186 | | | 0.004** | | | 0.019* |
| No formal or primary | 172 | 44 (25.6) | 128 (74.4) | | 20 (11.6) | 152 (88.4) | |
| Secondary or pre-university | 775 | 138 (17.8) | 637 (82.2) | | 62 (8.0) | 713 (92.0) | |
| University or higher | 239 | 31 (13.0) | 208 (87.0) | | 10 (4.2) | 229 (95.8) | |
| Menopausal status | 1201 | | | <0.001*** | | | 0.042* |
| Premenopausal | 151 | 17 (11.3) | 134 (88.7) | | 6 (4.0) | 145 (96.0) | |
| Perimenopausal | 189 | 19 (10.1) | 170 (89.9) | | 10 (5.3) | 179 (94.7) | |
| Postmenopausal | 861 | 180 (20.9) | 681 (79.1) | | 77 (8.9) | 784 (91.1) | |
| Smoking status | 1195 | | | 0.189 | | | 0.142 |
| Non-smoking | 1170 | 213 (18.2) | 957 (81.8) | | 93 (7.9) | 1077 (92.1) | |
| Smoking | 25 | 2 (8.0) | 23 (92.0) | | 0 (0.0) | 25 (100.0) | |
| Alcohol consumption | 1191 | | | 0.722 | | | 0.232 |
| No | 1153 | 208 (18.0) | 945 (82.0) | | 91 (7.9) | 1062 (92.1) | |
| Yes | 38 | 6 (15.8) | 32 (84.2) | | 1 (2.6) | 37 (97.4) | |
| Regular exercise | 1188 | | | 0.911 | | | 0.357 |
| Yes | 730 | 132 (18.1) | 598 (81.9) | | 53 (7.3) | 677 (92.7) | |
| No | 458 | 84 (18.3) | 374 (81.7) | | 40 (8.7) | 418 (91.3) | |
| Vitamin D (ng/ml) | 1192 | | | 0.161 | | | 0.162 |
| ≤ 20 | 241 | 36 (14.9) | 205 (85.1) | | 24 (10.0) | 217 (90.0) | |
| > 20 | 951 | 179 (18.8) | 772 (81.2) | | 69 (7.3) | 882 (92.7) | |
| Visceral adipose tissue (cm$^2$) | 1192 | | | <0.001*** | | | <0.001*** |
| ≤ 131 | 795 | 181 (22.8) | 614 (77.2) | | 45 (5.7) | 750 (94.3) | |
| > 131 | 397 | 33 (8.3) | 364 (91.7) | | 48 (12.1) | 349 (87.9) | |
| Hs-CRP (mg/L) | 1156 | | | 0.054 | | | <0.001*** |
| Lowest tertile | 405 | 83 (20.5) | 322 (79.5) | | 19 (4.7) | 386 (95.3) | |
| Middle tertile | 376 | 72 (19.1) | 304 (80.9) | | 21 (5.6) | 355 (94.4) | |
| Highest tertile | 375 | 53 (14.1) | 322 (85.9) | | 43 (11.5) | 332 (88.5) | |
| IL-6 (pg/mL) | 1196 | | | 0.314 | | | 0.014* |
| Lowest tertile | 589 | 114 (19.4) | 475 (80.6) | | 32 (5.4) | 557 (94.6) | |
| Middle tertile | 215 | 39 (18.1) | 176 (81.9) | | 20 (9.3) | 195 (90.7) | |
| Highest tertile | 392 | 61 (15.6) | 331 (84.4) | | 40 (10.2) | 352 (89.8) | |
| TNF-α (pg/mL) | 1200 | | | 0.201 | | | 0.004** |
| Lowest tertile | 413 | 71 (17.2) | 342 (82.8) | | 18 (4.4) | 395 (95.6) | |
| Middle tertile | 395 | 82 (20.8) | 313 (79.2) | | 34 (8.6) | 361 (91.4) | |
| Highest tertile | 392 | 63 (16.1) | 329 (83.9) | | 41 (10.5) | 351 (89.5) | |
| **Physical measures & performance** | | N (%) or mean ± SD | | | | | |
| ALM/height$^2$ (kg/m$^2$) | 1201 | | | <0.001*** | | | 0.692 |
| <5.4 | 622 | 216 (34.7) | 406 (65.3) | | 50 (8.0) | 572 (92.0) | |

(*Continued*)

**Table 1.** (Continued)

| Characteristics | Total n = 1201 | AWGS Sarcopenia n = 216 (18.0%) | AWGS No sarcopenia n = 985 (82.0%) | P-value | FNIH Sarcopenia n = 93 (7.7%) | FNIH No sarcopenia n = 1108 (92.3%) | P-value |
|---|---|---|---|---|---|---|---|
| Demographics | | | | Mean ± SD or N (%) | | | |
| ≥5.4 | 579 | 0 (0.0) | 579 (100.0) | | 43 (7.4) | 536 (92.6) | |
| ALM/BMI | 1201 | | | 0.004** | | | <0.001*** |
| <0.512 | 273 | 65 (23.8) | 208 (76.2) | | 93 (34.1) | 180 (65.9) | |
| ≥ 0.512 | 928 | 151 (16.3) | 777 (83.7) | | 0 (0.0) | 928 (100.0) | |
| Semi-tandem stand (s) | 1187 | | | 0.751 | | | 0.422 |
| <30 | 25 | 5 (20.0) | 20 (80.0) | | 3 (12.0) | 22 (88.0) | |
| 30 | 1162 | 204 (17.6) | 958 (82.4) | | 89 (7.7) | 1073 (92.3) | |
| Tandem stand (s) | 1201 | | | <0.001*** | | | <0.001*** |
| <30 | 217 | 67 (30.9) | 150 (69.1) | | 29 (13.4) | 188 (86.6) | |
| 30 | 984 | 149 (15.1) | 835 (84.9) | | 64 (6.5) | 920 (93.5) | |
| Maximum handgrip strength (kg) | 1201 | 13.3 ± 5.9 | 21.0 ± 5.5 | <0.001*** | 10.9 ± 3.0 | 20.3 ± 6.0 | <0.001*** |
| Repeated chair stand test (s) | 1180 | 12.6 ± 5.3 | 11.4 ± 3.4 | <0.001*** | 13.0 ± 5.0 | 11.5 ± 3.7 | <0.001*** |
| One-leg stand (s) | 1201 | 15.1 ± 8.1 | 16.5 ± 6.9 | 0.009** | 13.1 ± 8.3 | 16.6 ± 7.0 | <0.001*** |
| Gait speed usual walk (m/s) | 1189 | 0.96 ± 0.2 | 0.90 ± 0.2 | <0.001*** | 1.0 ± 0.3 | 0.9 ± 0.2 | <0.001*** |
| Gait speed narrow walk (m/s) | 1189 | 1.0 ± 0.3 | 0.9 ± 0.2 | <0.001*** | 1.0 ± 0.3 | 0.9 ± 0.2 | <0.001*** |
| Health conditions | | | | N (%) | | | |
| Hypertension | 1187 | | | 0.968 | | | 0.003** |
| No | 531 | 96 (18.1) | 435 (81.9) | | 28 (5.3) | 503 (94.7) | |
| Yes | 656 | 118 (18.0) | 538 (82.0) | | 65 (9.9) | 591 (90.1) | |
| Type 2 diabetes | 1189 | | | 0.393 | | | <0.001*** |
| No | 1047 | 193 (18.4) | 854 (81.6) | | 70 (6.7) | 977 (93.3) | |
| Yes | 142 | 22 (15.5) | 120 (84.5) | | 22 (15.5) | 120 (84.5) | |
| Osteoporosis | 1200 | | | 0.004** | | | 0.961 |
| No | 1124 | 193 (17.2) | 931 (82.8) | | 87 (7.7) | 1037 (92.3) | |
| Yes | 76 | 23 (30.3) | 53 (69.7) | | 6 (7.9) | 70 (92.1) | |
| Depression and/or anxiety symptoms | 1201 | | | 0.288 | | | 0.605 |
| No | 997 | 174 (17.5) | 823 (82.5) | | 79 (7.9) | 918 (92.1) | |
| Yes | 204 | 42 (20.6) | 162 (79.4) | | 14 (6.9) | 190 (93.1) | |
| Urinary incontinence | 1119 | | | 0.426 | | | 0.585 |
| No | 534 | 102 (19.1) | 432 (80.9) | | 40 (7.5) | 494 (92.5) | |
| Yes | 585 | 101 (17.3) | 484 (82.7) | | 49 (8.4) | 536 (91.6) | |

Values were presented as row percentages.

*p<0.05

**p<0.01

***p<0.001.

Results were analyzed using one-way ANOVA and Pearson's Chi -Square test. Missing data accounted for 0.1% to 6.8% of overall data.

## Discussion

In this cohort of midlife women, the prevalence of sarcopenia based on the AWGS criteria (18.0%) was higher than the FNIH criteria (7.7%). Women with osteoporosis were more likely to be sarcopenic by the AWGS criteria, while women with T2DM were more likely to be sarcopenic by the FNIH criteria, after adjustment for age, ethnicity, education, and menopause status.

Sarcopenia and key health conditions

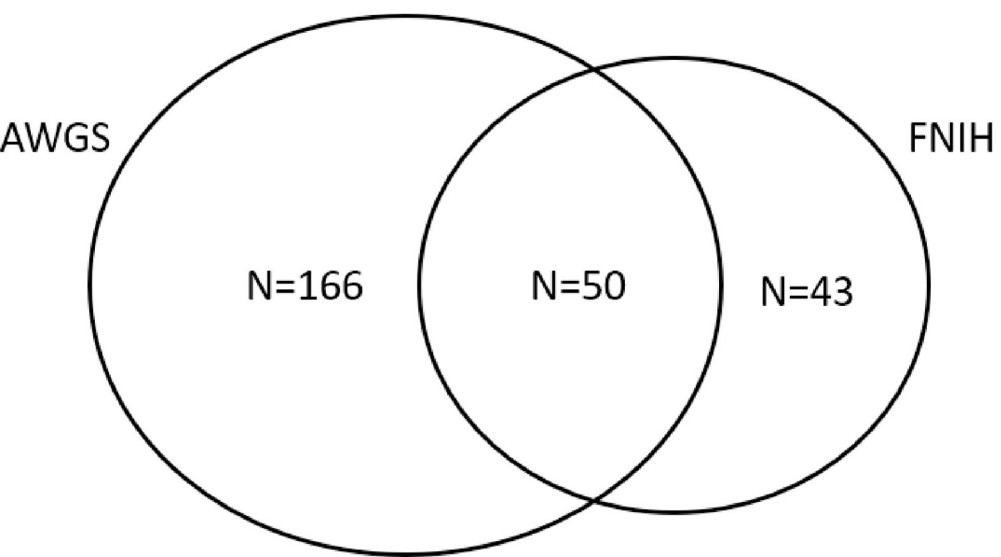

**Fig 1. Women defined as sarcopenic by AWGS (n = 216), FNIH (n = 93), and both criteria (n = 50).**

The prevalence of AWGS sarcopenia in our cohort (18.0%) and in another recent Singapore study (14.2% in women, aged 21–90 years) [12] was higher than that of other Asian populations in Taiwan (3.7% in women, mean age: 63.3 ± 10.0 years) [39] and Korea (13.8% in women, aged 70 to 84 years) [40]. A probable reason for the relatively higher sarcopenia prevalence in Singapore might be due to the warmer tropical climate, serving as a barrier to physical activity [41]. The well-developed transport infrastructure in Singapore might have reduced opportunities for active commuting, which is known to improve exercise capacity and cardiovascular outcomes [42].

To our knowledge, the FNIH definition has yet to be examined among the Singapore population. The FNIH prevalence of 7.7% concurs with findings reported in other Asian populations [43, 44]. Sarcopenic subjects, defined by FNIH, had greater VAT and higher levels of serum cardiometabolic markers than non-sarcopenic subjects. VAT is a risk factor for T2DM [45] and could possibly have explained the association between sarcopenia and T2DM in this study, especially since this association was attenuated upon VAT adjustment. Differences in associations with diabetes by the AWGS and FNIH may be due to variations in muscle mass indices used, namely, ALM/ht$^2$ and ALM/BMI. A Korean population-based study (n = 28,476) reported that the ASM/BMI index demonstrated an age-related decline at an onset of 30 years old, while the ASM/weight and ASM/height indices showed trends of a slight increase after 60 years and a steady increase until age 60 years respectively [46].

A recent systematic review reported that sarcopenia, defined by the earlier Asian and European definitions, was associated with hypertension, possibly through shared mechanisms of chronic inflammation and catabolic cytokines production [47]. Using the FNIH definition, we observed higher levels of inflammatory markers hs-CRP, TNF- α and IL-6 observed among sarcopenic women. However, this positive association was attenuated upon covariate adjustment, and more studies would be needed to elucidate the association between sarcopenia and hypertension. Osteoporosis at the lumbar spine was positively associated with sarcopenia before VAT adjustment. The confounding effect of VAT on sarcopenia observed in this study

**Table 2. Associations between health conditions and sarcopenia (n = 1201).**

| Health conditions | AWGS | FNIH |
|---|---|---|
| | OR (95% CI) | |
| Hypertension | | |
| Unadjusted | 1.00 (0.74, 1.35) | 2.10 (1.31, 3.36)** |
| Model 1 | 0.84 (0.61, 1.16) | 1.58 (0.97, 2.57) |
| Model 2 | 1.04 (0.75, 1.44) | 1.42 (0.86, 2.33) |
| Type 2 diabetes | | |
| Unadjusted | 0.84 (0.52, 1.35) | 2.64 (1.57, 4.44)*** |
| Model 1 | 0.81 (0.49, 1.34) | 1.98 (1.14, 3.42)* |
| Model 2 | 1.16 (0.68, 1.96) | 1.74 (0.99, 3.06) |
| Osteoporosis | | |
| Unadjusted | 2.10 (1.26, 3.52)** | 1.01 (0.43, 2.41) |
| Model 1 | 1.74 (1.02, 2.94)* | 0.81 (0.34, 1.96) |
| Model 2 | 1.36 (0.79, 2.33) | 0.95 (0.39, 2.32) |
| Depression and/or anxiety symptoms | | |
| Unadjusted | 1.23 (0.84, 1.80) | 0.80 (0.44, 1.47) |
| Model 1 | 1.26 (0.85, 1.87) | 0.80 (0.43, 1.49) |
| Model 2 | 1.23 (0.82, 1.83) | 0.82 (0.44, 1.53) |
| Urinary incontinence | | |
| Unadjusted | 0.90 (0.66, 1.22) | 1.23 (0.79, 1.92) |
| Model 1 | 0.93 (0.68, 1.28) | 1.21 (0.76, 1.90) |
| Model 2 | 1.03 (0.75, 1.43) | 1.14 (0.72, 1.81) |

*$P<0.05$

**$p<0.01$ and

***$p<0.001$. Results were analysed using binary logistic regression.

Model 1: Age, ethnicity, education level, menopausal status.

Model 2: Model 1 and VAT.

agrees with a meta-analysis of 44 studies [48] that found lean mass and fat mass to have similar effects on bone mineral density among postmenopausal women ($r = 0.33$ vs $r = 0.31$).

Sarcopenia was not associated with depression in our study, similar to findings from a Korean study (n = 7,364, women: 49.9%), that defined sarcopenia based on DXA-measured lean mass cut-offs from a younger reference population [49]. A meta-analysis among ~33,000 participants (aged >50 years, 56.3% women) reported that sarcopenia, measured by bioelectrical impedance analysis, was associated with depression compared to DXA-measured sarcopenia [50], highlighting the fact that different measuring tools might influence the association between sarcopenia and depression. The lack of a consistent measuring tool to define sarcopenia makes its association with depression problematic. Few studies have examined the relationship between urinary incontinence and sarcopenia. A recent study has reported that urinary incontinence (stress and/or urgency) was independently associated with low muscle mass adjusted by weight/BMI among older females (aged $\geq$ 60 years, n = 802) [51]. As there are different types and different methodologies to determine urinary incontinence, more studies would have to be conducted to ascertain the relationship between this condition and sarcopenia.

Physical activity reduces the odds of sarcopenia in later life (OR: 0.45, 95% CI: 0.37, 0.55) [52]. Findings from the Singapore National Health Survey 2020 indicate that a lower proportion of older adults exercised regularly compared to younger adults (29.2% vs 41.2%) [53].

Efforts to enhance physical activity in mid-life Singaporean women would need to address the reasons for low exercise levels which include time constraints due to work and family commitments and perceptions that housework provides adequate exercise [54].

The negative percent agreement between FNIH and AWGS was high at 95.6%, suggesting a good agreement in the absence of sarcopenia. Cases classified as non-sarcopenic by AWGS and FNIH were consistent and can possibly be relied upon as an indicator for the absence of muscle weakness. However, there was poor positive percent agreement of 23.1% between the AWGS and FNIH criteria, and $\kappa$ value was low at 0.24. Poor agreement between definitions is related to differences in thresholds used to define sarcopenia [55]. One key difference is the 2 kg higher grip strength cut-off in AWGS compared to FNIH, resulting in a higher proportion of subjects being defined as sarcopenic with the AWGS criteria. The AWGS hand grip strength criteria (<18 kg) represented the lowest quintile of several Asian cohorts [6] and was formulated to "facilitate frailty research in Asia." In contrast the FNIH criteria used classification and regression tree analysis to derive the grip strength cut-off (<16 kg), that was associated with 2.44-fold (95% CI 2.20–2.71) decrease in mobility impairment [56]. A challenge for sarcopenic research is to arrive at a consensus definition of sarcopenia, akin to the widespread acceptance of DXA-measured bone mineral density for osteoporosis [23]. Furthermore, the utility of novel tools to measure muscle mass such as the creatine dilution assay [57], or rapid MRI algorithms [58] need to be evaluated for their ability to generate consistent patterns of risk for muscle-related conditions as midlife women age.

A limitation of our study is the cross-sectional design, thereby precluding any conclusions as to causality or temporality. We are unable to ascertain if sarcopenia leads to T2DM and osteoporosis, or vice versa. However, longitudinal follow-up visits in our cohort are underway and will clarify the relationship between sarcopenia and incident health conditions as women age. We acknowledge that the criteria used in this study to define sarcopenia were derived from older populations and its application to our younger cohort might introduce inconsistencies. However, there is a need to study younger cohorts to identify manifestations of sarcopenia that present early, and these cut-offs have been recently used in other large cohort studies with a younger population [5, 59, 60]. Information on nutritional factors were not collected in the IWHP, However, we accounted for demographic and women-specific variables such as ethnicity, menopausal status, and VAT.

Strengths of this study include examining the relationship between sarcopenia and important health conditions prevalent among midlife women. Sarcopenia parameters were objectively measured and the questionnaires, although self-reported, were standardized and validated from large, established cohorts. The examination of several important health conditions in our study provides a holistic view of health among our women. The findings from our study are generalizable to the general population in Singapore as our participants' demographics were comparable to the rest of the population in terms of ethnicity (81% Chinese in our cohort vs 78% in the general population) and highest education level attained (20.2% university level qualification in our cohort vs 16.4% in the population) [61]. Our findings will also be generalizable to other midlife women in urbanized Asian cities.

## Conclusion

Sarcopenia was associated with T2DM and osteoporosis, suggesting that early interventions to maintain lean mass might prevent progression of these conditions. Even though they represent a single concept, sarcopenia by AWGS and FNIH criteria differed in their relationships with adverse health conditions. Rationalization of sarcopenia diagnostic criteria for more consistent ability to predict adverse health outcomes is needed.

## Supporting information

**S1 Table. Assay parameters for individual variables (n = 1201).**
(PDF)

## Acknowledgments

We would like to thank our IWHP study participants for availing themselves for this study. We would also like to thank our study coordinators, Ms. Jean Ho, Ms. Poon Peng Cheng, and Ms. Chua Seok Eng for helping with the data collection. We would like to thank Ms. Tan Sze Yee for performing the DXA scans and Dr. Li Jun for grant management and procurement of study materials.

## Author Contributions

**Conceptualization:** Susan Jane Sinclair Logan, Jane A. Cauley, Eu-Leong Yong.

**Formal analysis:** Beverly Wen-Xin Wong, Win Pa Pa Thu, Yiong Huak Chan.

**Funding acquisition:** Eu-Leong Yong.

**Methodology:** Eu-Leong Yong.

**Project administration:** Win Pa Pa Thu.

**Supervision:** Susan Jane Sinclair Logan, Jane A. Cauley, Eu-Leong Yong.

**Writing – original draft:** Beverly Wen-Xin Wong, Eu-Leong Yong.

**Writing – review & editing:** Beverly Wen-Xin Wong, Win Pa Pa Thu, Yiong Huak Chan, Susan Jane Sinclair Logan, Jane A. Cauley, Eu-Leong Yong.

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
