## [Decision Letter · Decision Letter 0]

26 Dec 2022

PONE-D-22-13016Association of sarcopenia with important health conditions among community-dwelling Asian women.PLOS ONE

Dear Dr. Yong,

Thank you for submitting your manuscript to PLOS ONE. After careful consideration, we feel that it has merit but does not fully meet PLOS ONE’s publication criteria as it currently stands. Therefore, we invite you to submit a revised version of the manuscript that addresses the points raised during the review process.

 Please submit your revised manuscript by Feb 09 2023 11:59PM. If you will need more time than this to complete your revisions, please reply to this message or contact the journal office at plosone@plos.org. Please include the following items when submitting your revised manuscript:A rebuttal letter that responds to each point raised by the academic editor and reviewer(s). You should upload this letter as a separate file labeled 'Response to Reviewers'.A marked-up copy of your manuscript that highlights changes made to the original version. You should upload this as a separate file labeled 'Revised Manuscript with Track Changes'.An unmarked version of your revised paper without tracked changes. You should upload this as a separate file labeled 'Manuscript'.

We look forward to receiving your revised manuscript.

Kind regards,

Kyung-Wan Baek, Ph.D.

Academic Editor

PLOS ONE

Journal Requirements:

“This study was partially funded by the Singapore National Medical Research Council Grant (Reference number: NMRC/CSA-SI/0010/2017) for ELY.”

4. We suggest you thoroughly copyedit your manuscript for language usage, spelling, and grammar. If you do not know anyone who can help you do this, you may wish to consider employing a professional scientific editing service.

****

Additional Editor Comments (if provided):

Although one reviewer submitted a "reject" opinion, two reviewers rated the quality of this manuscript highly. It is judged that if the manuscript is appropriately revised by considering the opinions of the reviewer who submitted the "reject" opinion, it can be published on PLoS One.

Please revise the manuscript based on the reviewer's opinion.

Reviewers' comments:

Reviewer's Responses to Questions

**Comments to the Author**

1. Is the manuscript technically sound, and do the data support the conclusions?

Reviewer #1: Yes

Reviewer #2: Yes

Reviewer #3: Yes

2. Has the statistical analysis been performed appropriately and rigorously? 

Reviewer #1: Yes

Reviewer #2: Yes

Reviewer #3: Yes

3. Have the authors made all data underlying the findings in their manuscript fully available?

Reviewer #1: Yes

Reviewer #2: Yes

Reviewer #3: Yes

4. Is the manuscript presented in an intelligible fashion and written in standard English?

Reviewer #1: Yes

Reviewer #2: Yes

Reviewer #3: Yes

5. Review Comments to the Author

Reviewer #1: This is a very well written and well presented manuscript. I had only one comment for the authors to consider:

Regarding the sentence: “Chinese women made up the majority 159 at 81.1% (Table 1), while Malay and Indian women constituted 5.5% and 9.9% respectively,” the percentages do not add up to 100%. It may be useful to state that in this sentence to help the reader understand that this is due to missing data.

Reviewer #2: General comment

First of all, thank you for giving me the opportunity to review this study.

The aim of this study was to investigate the prevalence of sarcopenia in midlife Singaporean women, using two different definitions of sarcopenia (Asian Working Group for Sarcopenia 2019 and Foundation for the National Institutes of Health). The study also examined the association of sarcopenia with various health conditions including hypertension, type 2 diabetes, osteoporosis, depression/anxiety, and urinary incontinence. The study included 1201 healthy community-dwelling women aged 45-69 years and measured muscle mass and function using dual-energy X-ray absorptiometry, handgrip strength, and the Short Physical Performance Battery. The results showed that the prevalence of sarcopenia was 18.0% and 7.7% according to the Asian Working Group for Sarcopenia and Foundation for the National Institutes of Health definitions, respectively. The study also found that sarcopenia was positively associated with osteoporosis and type 2 diabetes according to the Asian Working Group for Sarcopenia and Foundation for the National Institutes of Health definitions, respectively, but was not associated with hypertension. The study suggests that there may be differences in the relationship between sarcopenia and certain health conditions depending on the definition of sarcopenia used, and that further research is needed to rationalize diagnostic criteria for sarcopenia.

I believe that this type of research has been conducted so frequently that it does not provide particularly useful or novel information. Therefore, it would be more accurate to refer to this as a conceptual study that evaluates the relationship between specific diseases and potential new risk factors in a well-planned prospective cohort, rather than a study of the relationship between various health conditions.

Unfortunately, I believe that this study does not contribute significant new scientific knowledge to the Plosone journal, and therefore it may be difficult for it to progress through the publication process.

Specific comment

Method

Please explain the cohort definition in more detail.

Please draw and explain the flow chart for the subjects included in the final study in the study target group.

The average age of the study population is too young to study sarcopenia.

Reviewer #3: This is a good manuscript, well written, very clear, with a high sample, with a clear message and appropriate statistics. Unfortunately this is a cross sectional study and the association between sarcopenia and other pathologies remains difficult to interpret. Please find some comments below:

Introduction:

Authors report data of MA on falls and mortality. I know that there are other high-quality SR/MA reporting data over other outcomes of sarcopenia. Please cite some of them to offer a more accurate bibliography.

Why did the authors chosen the FNIH definition to be compared with AWGS? Why only reporting prevalence for those two particular definitions?

The secondary objectives of this paper (i.e. measuring associations of sarcopenia with hypertension, T2DM, osteoporosis, depression and anxiety) have not been introduced. Please provide a rationale for those analyses.

Method:

Please follow STROBE statement and report the use of this checklist in the methods section.

Did authors calibrated the DXA and Jamar ?

How did authors measured the normality of variables?

Results:

This is surprising to have such a high prevalence of sarcopenia in health community dwelling participants so young (45-69 years). Do you have an hypothesis to explain this high prevalence ? Could you also tell a bit more about the representativity of the sample?

Please report 95%CI for the Kappa agreement between both diagnoses. I would be interesting also to know how many among the 93 individuals diagnosed as sarcopenic using the FNIH criteria also are diagnosed sarcopenic with the AWGS criteria.

In Table 2, please remove the “adjusted” OR since some of them were crude OR.

6. PLOS authors have the option to publish the peer review history of their article (what does this mean?). If published, this will include your full peer review and any attached files.

Reviewer #1: No

Reviewer #2: No

Reviewer #3: No

---

## [Author Response · Author response to Decision Letter 0]

8 Jan 2023

RESPONSE TO EDITOR

We would like to amend our Funding Statement to now read: “This study was funded by the Singapore National Medical Research Council Grant (Reference number: NMRC/CSA-SI/0010/2017) for ELY. There was no additional external funding received for this study.” 

RESPONSE TO REVIEWER 1 

This is a very well written and well presented manuscript. I had only one comment for the authors to consider: Regarding the sentence: “Chinese women made up the majority 159 at 81.1% (Table 1), while Malay and Indian women constituted 5.5% and 9.9% respectively,” the percentages do not add up to 100%. It may be useful to state that in this sentence to help the reader understand that this is due to missing data.

Response: We thank reviewer 1 for his/her kind comments. We have added the following sentence in lines 189-190 of the tracked changes document: “The remaining 3.5% constituted of women from other ethnicities.” 

RESPONSE TO REVIEWER 2

I believe that this type of research has been conducted so frequently that it does not provide particularly useful or novel information.

Response: The novelty of our research is that we examined several important health conditions specific to midlife women to provide a holistic view of the relationship between these health conditions and sarcopenia. As mentioned in lines 33-39 of the tracked changes document: “In addition, most studies globally (8, 9) and in Singapore (14, 15) focused mainly on the relationship between sarcopenia and single health conditions. In this respect, the Integrated Women’s Health Program (IWHP) has identified important health conditions that manifest in midlife women, including high systolic blood pressure (20), insulin resistance (21), osteoporosis (22), depression and anxiety (23), and urinary incontinence (24).” 

Most sarcopenia studies were conducted in Caucasian populations and did not focus on midlife women. We found it particularly valuable to conduct such studies in midlife women as mentioned in lines 29-32: “and there is some evidence that muscle mass reduction accelerates during the time around menopause (18). Decrease in estrogen levels post-menopause have been associated with a decrease in muscle mass and function (18, 19).”

Therefore, it would be more accurate to refer to this as a conceptual study that evaluates the relationship between specific diseases and potential new risk factors in a well-planned prospective cohort, rather than a study of the relationship between various health conditions.

Response: Rather than being conceptual, this paper presents empirical data to examine the relationship of sarcopenia, as defined by AWGS and FNIH, with important health conditions affecting midlife women using objectively measured parameters and validated questionnaires. 

Please explain the cohort definition in more detail.

Response: To further define the cohort, we have included the following in lines 54-70: “The cohort has been fully described previously (25). Briefly, healthy women aged 45-69 years receiving routine gynecological follow-up at well-women clinics at the National University Hospital were recruited from September 2014 to October 2016 through fliers, posters, and word-of-mouth. Exclusion criteria included pregnancy or being severely ill. The number of women screened was at 2715, and 2191 met the eligibility criteria. Thereafter, 746 subjects, 134 subjects and 110 subjects declined participation, failed to attend the appointment and were non-contactable respectively, leaving a final sample of 1201 women (25). The protocol was approved by the Domain Specific Review Board of the National Healthcare Group, Singapore (Reference number: 2014/00356) and all participants provided written informed consent. In short, a whole-body composition scan was performed, and a fasting blood sample was collected for participants on arrival. This was followed by light refreshments, a series of questionnaires, biophysical measurements, and physical function tests, with the visit approximating 90 minutes.” 

Please draw and explain the flow chart for the subjects included in the final study in the study target group.

Response: The flowchart for the current cohort has been previously published in reference (25). To give a clearer overview of subjects enrolled in this study, we have additionally included the following sentence in lines 59-63: “The number of women screened was at 2715, and 2191 met the eligibility criteria. Thereafter, 746 subjects, 134 subjects and 110 subjects declined participation, failed to attend the appointment and were non-contactable respectively, leaving a final sample of 1201 women.” 

The average age of the study population is too young to study sarcopenia.

Response: In lines 25-26, we have acknowledged that “studies on sarcopenia were mostly conducted in elderly subjects.” Nevertheless, this study focuses on midlife subjects for the following reasons mentioned in lines 26-33: “Nevertheless there is increasing realization that the development of sarcopenia may start earlier in life (16), emphasizing the need to study sarcopenia across the life course. Women have lower absolute muscle mass than men (17), and there is some evidence that muscle mass reduction accelerates during the time around menopause (18). Decrease in estrogen levels post-menopause have been associated with a decrease in muscle mass and function (18, 19). Hence, studies on sarcopenia should also focus on younger women as much as in geriatric men.”

RESPONSE TO REVIEWER 3

This is a good manuscript, well written, very clear, with a high sample, with a clear message and appropriate statistics. Unfortunately this is a cross sectional study and the association between sarcopenia and other pathologies remains difficult to interpret. 

Response: We thank the reviewer for his/her kind comments about the manuscript. We agree that this cross-sectional study would not be able to dissect the cause-and-effect relationship between sarcopenia and other health conditions, and this has been mentioned from lines 336-338 of the tracked changes document: “A limitation of our study is the cross-sectional design, thereby precluding any conclusions as to causality or temporality. We are unable to ascertain if sarcopenia leads to T2DM and osteoporosis, or vice versa.” 

To address this limitation, we have almost completed our 7-year follow-up of the cohort, and the conclusions of this follow-up study would be published soon, as stated in lines 338-341, “However, longitudinal follow-up visits in our cohort are underway and will clarify the relationship between sarcopenia and incident health conditions as women age.” 

Authors report data of MA on falls and mortality. I know that there are other high-quality SR/MA reporting data over other outcomes of sarcopenia. Please cite some of them to offer a more accurate bibliography.

Response: We have included “functional decline, reduced quality of life” and “cardiovascular diseases” as other outcomes of sarcopenia, citing three additional papers from high-quality journals for a more accurate bibliography. Lines 2-5 state that: “Sarcopenia, the loss of skeletal muscle mass with advancing years, is associated with increased risk of falls (1), functional decline, reduced quality of life (2), higher hospitalization rates and related costs (3), cardiovascular diseases (4), disability (5), and even mortality (5, 6).”

Why did the authors chosen the FNIH definition to be compared with AWGS? Why only reporting prevalence for those two particular definitions? 

Response: We agree that there are many definitions of sarcopenia. We chose the AWGS as it is most appropriate for Asian women. Furthermore, the AWGS definition is similar to, being a derivative of the European Working Group on Sarcopenia in Older People (EWGSOP). We included the FNIH definition as it is commonly used by sarcopenia researchers globally. 

The secondary objectives of this paper (i.e. measuring associations of sarcopenia with hypertension, T2DM, osteoporosis, depression and anxiety) have not been introduced. Please provide a rationale for those analyses.

Response: To make the secondary objectives of the paper clearer, we have mentioned the following in lines 35-45: “In this respect, the Integrated Women’s Health Program (IWHP) has identified important health conditions that manifest in midlife women, including high systolic blood pressure (20), insulin resistance (21), osteoporosis (22), depression and anxiety (23), and urinary incontinence (24). Since hypertension, T2DM, osteoporosis, depression and anxiety, and urinary incontinence are the most common and critical health conditions, we studied their association with sarcopenia as these conditions are likely to progress and result in a higher healthcare burden on midlife women as they age. Identification of the link between sarcopenia and these common health conditions would allow for specific preventive healthcare initiatives.”

Method: Please follow STROBE statement and report the use of this checklist in the methods section.

Response: The design of our study follows all the items in the STROBE statement, and this is now indicated in line 74-75: “This study was performed in concordance with all the items required in the STROBE statement (26).”

Did authors calibrated the DXA and Jamar?

Response: Yes, we calibrated the DXA and Jamar on a regular basis. We have added the following information in lines 110-114: “Operators of the DXA machine were trained to follow standard protocols according to the manufacturer’s instructions. The DXA machine was calibrated daily and sent for maintenance every 6 months. Handgrip strength was assessed using a dynamometer (Jamar, Bolingbrook, IL), which was calibrated yearly.”

How did authors measured the normality of variables?

Response: We have added the information in lines 160-161: “The normality of the variables was examined visually using histograms and Q-Q plots, as well as using the Kolmogorov-Smirnov and Shapiro-Wilk tests.”

This is surprising to have such a high prevalence of sarcopenia in health community dwelling participants so young (45-69 years). Do you have an hypothesis to explain this high prevalence? 

Response: 

We have now included the following reasons to possibly explain the high prevalence of sarcopenia in our young participants in lines 253-262: “The prevalence of AWGS sarcopenia in our cohort (18.0%) and in another recent Singapore study (14.2% in women, aged 21-90 years) (12) was higher than that of other Asian populations in Taiwan (3.7% in women, mean age: 63.3 ± 10.0 years) (39) and Korea (13.8% in women, aged 70 to 84 years) (40). A probable reason for the relatively higher sarcopenia prevalence in Singapore might be due to the warmer tropical climate, serving as a barrier to physical activity (41). The well-developed transport infrastructure in Singapore might have reduced opportunities for active commuting, which is known to improve exercise capacity and cardiovascular outcomes (42).” 

Could you also tell a bit more about the representativity of the sample?

Response: Our cohort closely reflects the demographics of women in Singapore and we have added that in lines 356-362: “The findings from our study are generalizable to the general population in Singapore as our participants’ demographics were comparable to the rest of the population in terms of ethnicity (81% Chinese in our cohort vs 78% in the general population) and highest education level attained (20.2% university level qualification in our cohort vs 16.4% in the population) (61). Our findings will also be generalizable to other midlife women in urbanized Asian cities.”

I would be interesting also to know how many among the 93 individuals diagnosed as sarcopenic using the FNIH criteria also are diagnosed sarcopenic with the AWGS criteria. 

Response: In the Venn diagram (Figure 1), we have indicated that out of 93 individuals diagnosed as sarcopenic using the FNIH criteria, n=50 were also diagnosed as sarcopenic with the AWGS criteria as seen from the overlapped region between the two circles. 

Please report 95%CI for the Kappa agreement between both diagnoses.

Response: We have reported the 95% CI for the Kappa agreement in line 195: “Cohen’s � of 0.24 (95%CI: 0.17, 0.31) indicated poor agreement between the two criteria.” 

In Table 2, please remove the “adjusted” OR since some of them were crude OR.

Response: We have removed the word “adjusted” in table 2.

---

## [Decision Letter · Decision Letter 1]

16 Jan 2023

Association of sarcopenia with important health conditions among community-dwelling Asian women.

PONE-D-22-13016R1

Dear Dr. Yong,

We’re pleased to inform you that your manuscript has been judged scientifically suitable for publication and will be formally accepted for publication once it meets all outstanding technical requirements.

Kind regards,

Kyung-Wan Baek, Ph.D.

Academic Editor

PLOS ONE

Additional Editor Comments (optional):

Congratulations. The authors collected the reviewers' opinions and revised the manuscript to be suitable for publication in PLoS one. I approve this publication.

Reviewers' comments:

Reviewer's Responses to Questions

**Comments to the Author**

1. If the authors have adequately addressed your comments raised in a previous round of review and you feel that this manuscript is now acceptable for publication, you may indicate that here to bypass the “Comments to the Author” section, enter your conflict of interest statement in the “Confidential to Editor” section, and submit your "Accept" recommendation.

Reviewer #1: All comments have been addressed

Reviewer #3: All comments have been addressed

2. Is the manuscript technically sound, and do the data support the conclusions?

Reviewer #1: Yes

Reviewer #3: Yes

3. Has the statistical analysis been performed appropriately and rigorously? 

Reviewer #1: Yes

Reviewer #3: Yes

4. Have the authors made all data underlying the findings in their manuscript fully available?

Reviewer #1: Yes

Reviewer #3: Yes

5. Is the manuscript presented in an intelligible fashion and written in standard English?

Reviewer #1: Yes

Reviewer #3: Yes

6. Review Comments to the Author

Reviewer #1: The authors have done well responding to the reviewers’ comments. I am happy to recommend this manuscript for publication.

Reviewer #3: (No Response)

7. PLOS authors have the option to publish the peer review history of their article (what does this mean?). If published, this will include your full peer review and any attached files.

Reviewer #1: No

Reviewer #3: No

---

## [Editor Report · Acceptance letter]

20 Jan 2023

PONE-D-22-13016R1 

Association of sarcopenia with important health conditions among community-dwelling Asian women. 

Dear Dr. Yong:

I'm pleased to inform you that your manuscript has been deemed suitable for publication in PLOS ONE. Congratulations! Your manuscript is now with our production department. 

Kind regards, 

on behalf of

Dr. Kyung-Wan Baek 

Academic Editor

PLOS ONE